# Identification, Pathogenicity, and Sensitivity to Fungicide of *Colletotrichum* Species That Causes Walnut Anthracnose in Beijing

**Fuxin Li** [1,†], **Jiawen Chen** [1,†], **Qian Chen** [2], **Ziyi Liu** [1], **Junyuan Sun** [1], **Yitong Yan** [1], **Hanxing Zhang** [3] **and Yang Bi** [1,*]

1   Key Laboratory for Northern Urban Agriculture of Ministry of Agriculture and Rural Affairs, Beijing University of Agriculture, Beijing 102206, China
2   State Key Laboratory of Mycology, Institute of Microbiology, Chinese Academy of Sciences, Beijing 100101, China
3   Key Laboratory of Systematic Mycology and Lichenology, Institute of Microbiology, Chinese Academy of Sciences, Beijing 100080, China
*   Correspondence: biyang0620@126.com
†   These authors contributed equally to this work.

**Abstract:** Walnuts (*Juglans regia* L.) are a major food crop in Beijing. Recently, walnut anthracnose has become a serious problem in walnut plantations of Beijing, and the diversity of pathogens that cause this disease is poorly understood, making targeted treatment difficult. This study investigated incidences of walnut anthracnose in seven districts of Beijing (Changping, Fangshan, Haidian, Huairou, Mentougou, Miyun and Pinggu). A total of 377 isolates of *Colletotrichum* spp. were obtained from walnut anthracnose infected leaves and fruits. Morphological observation and multigene phylogenetic analysis of the genes ACT, CAL, CHS-1, GAPDH, ITS and TUB2 revealed that the isolates consisted of six species, *C. aenigma*, *C. fructicola*, *C. gloeosporioides*, *C. siamense*, *C. liaoningense* and *C. sojae*. Among these, *C. gloeosporioides* was the dominant species, and, for the first time, *C. liaoningense* and *C. sojae* were found to cause anthracnose in walnuts. Sensitivity tests showed that prochloraz and SYP-14288 had the strongest inhibitory effect on mycelial growth. These findings have clarified the species that cause walnut anthracnose in these seven districts of Beijing, which provides a scientific basis for diagnosis and control of walnut anthracnose.

**Keywords:** walnut anthracnose; polygenetic phylogeny; pathogenicity; fungicide; sensitivity test

## 1. Introduction

China is the largest walnut (*Juglans regia* L.) producing country in the world, yielding half of the world's total output [1]. According to the latest FAO statistics, China produced nearly 1.6 million tons of walnuts on more than 390,000 hectares, including 14,000 hectares in Beijing. Many fungi can infect walnuts causing anthracnose, leaf spot, powdery mildew and ulcers [2–5]. Among them, anthracnose is the most serious; it has a long incubation period and concentrated onset time, which results in strong outbreaks.

Anthracnose infection of walnut can damage fruits, leaves and branches. The symptoms on the fruit include round brown spots in the early stage that later turn black. The centers of the spots are covered with small black spots. As environmental humidity increases, orange-red conidia appear in a whorled arrangement. As the disease worsens, the number and size of disease spots on the fruit increase. The disease spots gradually expand and connect, causing the whole fruit to blacken and rot, resulting in fruit drop [6]. After the leaves are infected by anthracnose, they often form nearly round or irregular black or brown spots and gradually wither.

Walnut anthracnose is caused by *Colletotrichum* spp., which is recognized as the eighth most important plant pathogen worldwide because of its significant impact on agricultural

and forestry production [7]. Its asexual form belongs to *Colletotrichum* and its sexual form belongs to *Glomerella*. *Colletotrichum* spp., such as *C. acutatum*, *C. kahawae*, *C. gloeosporioides* and *C. graminicola*, also infect the fruit of other plants, including strawberry, blueberry, kiwi, coffee, persimmon, apple, grape, pear and others [8–13], reducing production to varying degrees.

*Colletotrichum* was first observed by Tode in 1790, and it was described in detail by Corda in 1831. With the development of multigene phylogeny, Cai et al. proposed a phylogenetic approach to compare isolates of *Colletotrichum* to type strains that also includes morphology, physiology, pathogenicity, culture characteristics and secondary metabolites [14]. As a result, the systematics of *Colletotrichum* have changed significantly and have had several revisions as many new species have been identified and typified. As new species were added, some were reclassified, combined or abolished. By 2019, a total of 213 *Colletotrichum* species were recognized [15–25].

Studies have shown that *Colletotrichum* spp. may cause walnut anthracnose in China, and the pathogen was identified as *C. gloeosporioides*, based on morphologic characters and ITS data analysis [26]. Soon after, *C. gloeosporioides* was found to be a species complex that included 38 related species. Classifying only by morphology or ITS sequences failed to justify which species comprised the complex. Several mycologists who worked on *Colletotrichum* classification complained that the data used to construct the prevailing taxonomy system of *Colletotrichum* (i.e., morphological characteristics, host range and ITS) were inconsistent [27]. Therefore, new approaches to accurately distinguish the species causing each infection are critical to understand and treat the infections.

Walnut anthracnose has been studied in major walnut producing areas, including Shandong, Sichuan, Xinjiang, Hebei and Guangxi. In those studies, *C. gloeosporioides* species complex was implicated in walnut anthracnose infection events [28–30]. In recent studies, *C. aenigma* [31]; *C. acutatum*, *C. gloeosporioides* and *C. fioriniae* [32]; *C. fructicola* [33]; and *C. siamense* and *C. viniferum* [34] have been reported to cause anthracnose in walnuts. However, the pathogen composition of walnut anthracnose in Beijing was still undefined.

In this study, field investigations were carried out in walnut orchards in seven regions of Beijing to clarify the type, occurrence and severity of walnut anthracnose. We identified the dominant walnut anthracnose species, their pathogenicity as well as their sensitivity to different fungicides. The results provide a theoretical basis for the scientific and rational use of fungicides to treat anthracnose in walnut plantations in various districts of Beijing.

## 2. Materials and Methods

Plant collection and field investigations. Walnut leaves and fruits with anthracnose symptoms were collected in seven districts (Changping, Fangshan, Haidian, Huairou, Mentougou, Miyun and Pinggu) of Beijing. Field investigations of walnut anthracnose were conducted in a walnut orchard in th4e Fangshan district of Beijing (June to September, 2020). We randomly selected 30 walnut trees in the Fangshan district with similar growth, then observed and recorded the symptoms in walnut leaves and fruits every month. At the same time, we calculated the disease incidence rate and disease index using a grading scale described in Tables S1 and S2 [35].

$$\text{Disease incident rate (\%)} = \frac{\text{diseased leaves or fruits}}{\text{total investigated leaves or fruits}} \times 100\%$$

$$\text{Disease index} = \frac{\sum \text{diseased leaves or fruits at each level} \times \text{representative}}{\text{total investigated leaves or fruits} \times \text{highest representative}} \times 100$$

Pathogen isolation and purification. Pure cultures were collected from diseased walnut leaves and fruits by single spore isolation [35]. The cleaned walnut leaves and fruits were cut into 4 mm$^2$ to 5 mm$^2$ segments with disease health junctions. The segments were successively soaked in 1% NaClO for 30 s, 75% ethanol for 30 s, and washed in sterile water three times. Then, segments were incubated on potato dextrose agar (PDA) plates at 28 °C

in the dark. Five days later, single spores were selected and transferred for purification, and isolated pathogens were obtained. Isolates were stored at 4 °C on PDA slants. All isolates were stored and maintained at Beijing University of Agriculture.

DNA extraction, amplification and sequencing. Isolates were incubated on PDA plates for 5 days and mycelia were collected in sterile centrifuge tubes. The appropriate zirconium beads (1 mm, 3 mm or 5 mm) were added and mycelia were triturated for 30 s three times. Genomic DNA was extracted with a DNA extraction kit (provided by AIBOSEN BIO, Beijing, China). PCR amplification of ribosomal internal transcribed spacer (ITS), actin (ACT), calmodulin (CAL), chitin synthase 1 (CHS-1), glyceraldehyde-3-phosphate dehydrogenase (GAPDH) and β-tubulin (TUB2) was conducted. Six loci were amplified: first using universal primers ITS-1 and ITS-4 for primary ascertainment of classification status [36], and then ACT gene using primers ACT-512F and ACT-783R [37]; CAL gene using primers CL1C and CL2C [38]; CHS-1 gene using primers CHS-79F and CHS-345R [37]; GAPDH genes using primers GDF1 and GDR1 [39]; and TUB2 gene using primers T1 and Bt2b [40,41]. Primer sequences, amplification size and PCR conditions are shown in Table 1.

**Table 1.** Primer information used in this study and PCR reaction procedure.

| Target | Primer | Sequence (5′ to 3′) | Direct | Initial Denaturation | | Denaturation | | Annealing | | Extension | | Final Extension | |
|---|---|---|---|---|---|---|---|---|---|---|---|---|---|
| | | | | °C | Sec | °C | Sec | °C | Sec | °C | Sec | °C | Sec |
| ACT | ACT-512F | ATG TGC AAG GCC GGT TTC GC | F | 94 | 120 | 94 | 10 | 58 | 10 | 72 | 60 | 72 | 300 |
| | ACT-783R | TAC GAG TCC TTC TGG CCC AT | R | 94 | 120 | 94 | 10 | 58 | 10 | 72 | 60 | 72 | 300 |
| CAL | CL1C | GAA TTC AAG GAG GCC TTC TC | F | 94 | 120 | 94 | 10 | 55 | 10 | 72 | 60 | 72 | 300 |
| | CL2C | CTT CTG CAT I GAG CTG GAC | R | 94 | 120 | 94 | 10 | 55 | 10 | 72 | 60 | 72 | 300 |
| CHS-1 | CHS-79F | TGG GGC AAG GAT GCT TGG AAG AAG | F | 94 | 120 | 94 | 10 | 58 | 10 | 72 | 60 | 72 | 300 |
| | CHS-345R | TGG AAG AAC CAT CTG TGA GAG TTG | R | 94 | 120 | 94 | 10 | 58 | 10 | 72 | 60 | 72 | 300 |
| GAPDH | GDF1 | GCC GTC AAC GAC CCC TTC ATT GA | F | 94 | 120 | 94 | 10 | 60 | 10 | 72 | 60 | 72 | 300 |
| | GDR1 | GGG TGG AGT CGT ACT GCA TGT | R | 94 | 120 | 94 | 10 | 60 | 10 | 72 | 60 | 72 | 300 |
| ITS | ITS-1 | TCC GTA GGT GAA CCT GCG G | F | 94 | 120 | 94 | 10 | 55 | 10 | 72 | 60 | 72 | 300 |
| | ITS-4 | TCC Gerber GCT TAT TGA TAT GC | R | 94 | 120 | 94 | 10 | 55 | 10 | 72 | 60 | 72 | 300 |
| TUB2 | T1 | AAC ATG CGT GAG ATT GTA AGT | F | 94 | 120 | 94 | 10 | 55 | 10 | 72 | 60 | 72 | 300 |
| | Bt2b | ACC CTC AGT GTA GTG ACC CTT GGC | R | 94 | 120 | 94 | 10 | 55 | 10 | 72 | 60 | 72 | 300 |

The PCR system consisted of 50 μL containing 46 μL 1×Taq MasterMix (Taq DNA polymerase and ddH$_2$O), primer-F (1.0 μL), primer-R (1.0 μL) and genomic DNA (2.0 μL). The PCR amplification procedure consisted of 35 cycles. The information about initial denaturation, denaturation, annealing, extension and final extension are listed in Table 1. After amplification, agarose gel electrophoresis was performed using 1% agarose gel and 10 μL GelRed nucleic acid dye to test bands. All above PCR reagents were provided by Beijing Biomed Gene Technology Co., Ltd., Beijing, China. PCR products with detected bright bands were sequenced by Beijing Biomed Gene Technology Co., Ltd.

Phylogenetic analysis. Related species sequences were downloaded from GenBank at NCBI (https://www.ncbi.nlm.nih.gov/, accessed on 14 November 2020) as model sequences (Table S3). Nucleotide sequences of the sequenced and reference strains were merged and collated using Mega v.7.0.26 [42], further aligned using MAFFT v.7 and manually corrected [43]. Six sequences were concatenated manually using Mega v.7.0.26 in the order of ACT, CAL, CHS-1, GAPDH, ITS and TUB2. For better discrimination, four columns of "N" were added between

each sequence. Phylogenetic analyses were performed with the concatenated sequences using MEGA v.7.0.26. Phylogenetic trees were constructed by the Mrbayes method [44]. All sequences have been uploaded to the GenBank database at NCBI.

Morphological observation. Isolates inoculated for observation were cultured on PDA plates at 28 °C for 5 days. Each isolate was cultured in three replicate plates. The colony size, color and morphology were measured and recorded daily. Mycelia were picked for observation under an optical microscope. After sporulation, the spore suspension was prepared by picking up the conidia with sterile water in a centrifuge tube and vortex oscillation. A few drops were placed on a glass slide and gently covered with glass. Conidial morphology was observed under an optical microscope.

Temperature effect. Isolates from different regions and species were selected to determine the effect of temperature on colony growth. The temperature settings were: 5, 10, 15, 20, 25, 28, 30, 32, 32, 34, 36, and 38 °C. The colony diameter (mm) was measured after 4 days and the daily growth rate (mm/d) was calculated. Each treatment was replicated three times and the experiment was replicated twice.

Pathogenicity and virulence of detached leaves. Fresh walnut leaves of similar growth were collected from walnut trees in the Fangshan district. Conidial suspensions ($10^6$ spores/mL) of pathogens were used for stab inoculation of detached leaves [35]. First, the leaf surface was moistened with sterile water for 30 s, soaked with 75% ethanol for 30 s, washed three times with sterile water and air dried. After that, the wound was stabbed through the veins on each side of the leaf with a 0.5 mm inoculation needle. Eight μL of conidial suspension was inoculated on the right side of leaf veins and an equal volume of sterile water was injected on the left side as the control. All treatments were incubated in a constant temperature artificial climate box at 26 °C with 80% humidity, 12 h light and 12 h dark for 5 days. A diameter cross measurement was used to record the lesion size. The inoculation point was the center, and cross diameters were measured separately. The final lesion size was reported as the average of all measurements. Inoculation of each strain was repeated three times and the experiments were performed twice.

In vitro sensitivity of isolates to seven fungicides. The inhibitive activities of prochloraz, difenoconazole, SYP-14288, epoxiconazole, fluazinam, mefentrifluconazole and tebuconazole of the isolates were estimated using a mycelium growth rate method.

Prochloraz (97% a.i., provided by Hangzhou Qingfeng agrochemical Co., Ltd., Zhejiang, China), difenoconazole (95.8% a.i., provided by Hangzhou Yulong agrochemical Co., Ltd., Zhejiang, China), fluazinam (97.5% a.i.), SYP-14288 (92% a.i., provided by Shenyang Research Institute of Chemical Industry, Liaoning, China), epoxiconazole (97.5% a.i., provided by Henan Zhongyuan seed coating agent factory, Henan, China), mefentrifluconazole (97.5% a.i., provided by Shenyang Institute of Chemical technology, Liaoning, China) and tebuconazole (97.5% a.i., provided by Jiangsu Fengdeng pesticide Co., Ltd., Jiangsu, China) were dissolved in dimethyl sulfoxide (DMSO) to make 10 mg/mL solutions. All fungicides were serially diluted as follows: difenoconazole (0.02, 0.05, 0.1, 0.5, 1 mg/mL), fluazinam (0.01, 0.03, 0.05, 0.1, 0.2 mg/mL), epoxiconazole (0.1, 0.2, 0.4, 1, 2 mg/mL), mefentrifluconazole (0.2, 0.5, 1, 2, 4 mg/mL), prochloraz (0.005, 0.01, 0.02, 0.04, 0.1 mg/mL), SYP-14288 (0.004, 0.01, 0.02, 0.05, 0.1 mg/mL) and tebuconazole (0.05, 0.1, 0.2, 0.5, 1, 2 mg/mL).

Seven fungicides were added to sterilized PDA at a ratio of 1:1000. PDA medium with DMSO was used as the control group. Approximately 15 mL PDA medium was poured into each plate, and isolates were cultured at the center of PDA plates at 28 °C for 3 days. Each treatment and control had three replicates and trials were performed twice. The colony diameters were measured by the cross method. Percentage inhibition was calculated by the following formula:

$$\text{Inhibition (\%)} = \frac{\text{control colony diameter} - \text{treatment colony diameter}}{\text{control colony diameter}} \times 100$$

The toxicity regression equation was described as y = ax + b, where x was the log transformed fungicide concentration and y was the percent inhibition. The correlation

coefficient (r) was obtained according to the relationship between x and y, and the $EC_{50}$ (µg/mL) of the fungicide was calculated according to the regression equation. Statistical analysis of the data was performed with IBM SPSS Statistics 18.0 software. ANOVA with comparison of means ($p \leq 0.05$) and ANOVA with least significant difference ($p = 0.05$) were used to determine whether significant differences in $EC_{50}$ existed among groups.

## 3. Results

### 3.1. Field Investigations and Description of Symptoms

We investigated a walnut anthracnose incident that occurred from July to September in a plantation in the Fangshan district (N: 39°81′95.2″ and E: 116°04′09.0″) of Beijing. The investigated walnut trees were six years old, 4–6 m high and 10–20 cm in trunk diameter. The survey showed that walnut trees were growing healthily and no disease was found on the fruits and branches in early July (Figure 1a). In later July, anthracnose infected walnut leaves had spots that were nearly circular, water soaked and rounded with dark brown margins (Figure 1b). By the end of August, the disease had spread to the fruit and to more leaves (Figure 1c). In September, a large number of the fruits were black and rotten, and some had fallen off (Figure 1d). The symptoms of anthracnose on walnut fruits began with sporadically distributed black spots that grew in number (Figure 2a), eventually leading to dark brown necrotic lesions that were decaying internally (Figure 2b). On sufficiently humid days, pale pink conidia stacks arranged in concentric circles emerged (Figure 2c). Walnut leaves became chlorotic with dark brown edges and contained small, round lesions (Figure 2d). The brown lesions eventually developed a necrotic center (Figure 2e). Some had dark brown lesions, with wheel markings and brown edges (Figure 2f). A total of 900 walnut leaves and 300 fruits were observed in field investigations, 30 leaves and approximately 10 fruits each tree. The incidence rate increased from August to September, reaching 41.44% on walnut leaves and 45.43% on walnut fruits (Table 2).

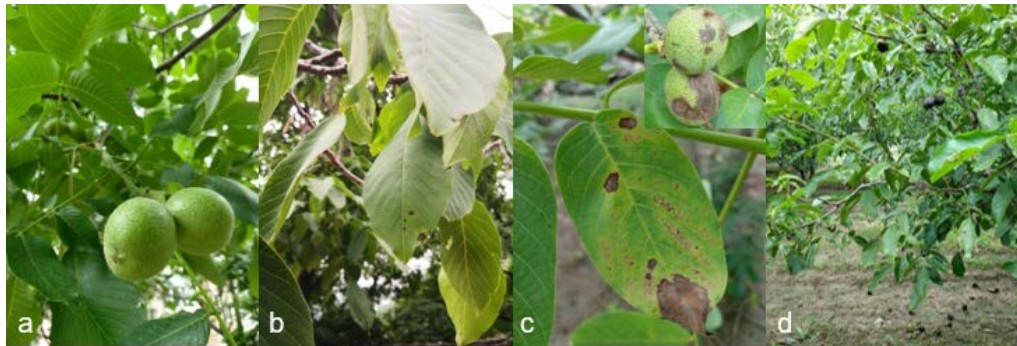

**Figure 1.** Disease symptoms at different stages of walnut anthracnose infection of leaves and fruits in the field. (**a**) 1 June (30 °C, 25% RH); (**b**) 11 July (27 °C, 90% RH); (**c**) 10 August (31 °C, 62% RH); (**d**) 15 September (24 °C, 93% RH).

**Table 2.** Disease incident rate and disease index of walnut anthracnose each month.

| Month | Part | Disease Incidence Rate (%) | Disease Index |
|:---:|:---:|:---:|:---:|
| June | Fruit | 0 | 0 |
| | Leaf | 0 | 0 |
| July | Fruit | 10.93 | 3.85 |
| | Leaf | 15.1 | 4.84 |
| August | Fruit | 24.53 | 26.7 |
| | Leaf | 12.21 | 13.52 |
| September | Fruit | 45.43 | 41.07 |
| | Leaf | 41.44 | 27.32 |

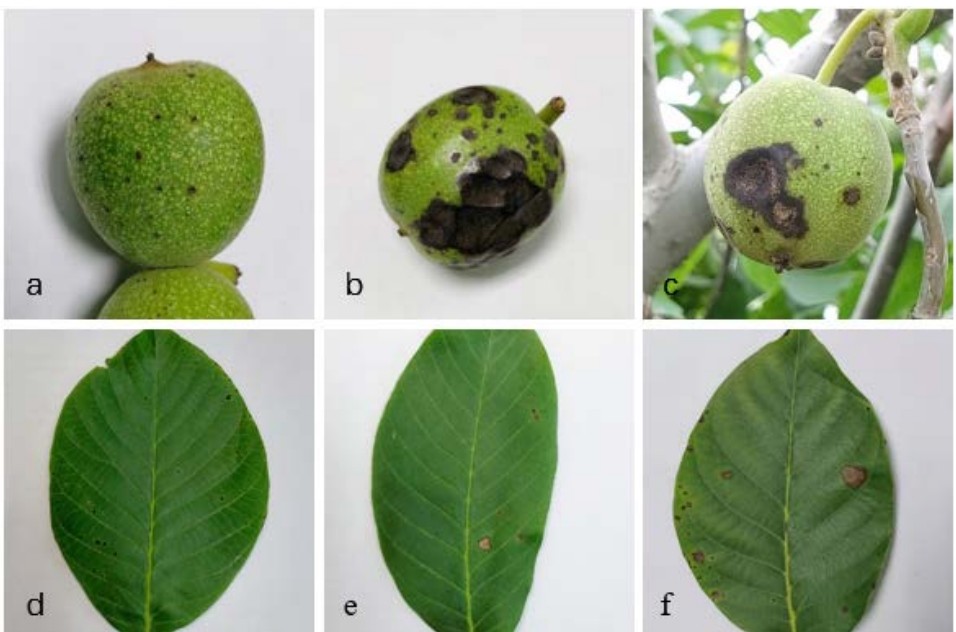

**Figure 2.** Symptoms of walnut anthracnose on fruits and leaves. (**a**) small black spots on fruits (diameter: <2 mm); (**b**) black sunken rot lesions on fruits (diameter: 5–35 mm); (**c**) brown necrotic lesions on fruits (diameter: 5–35 mm); (**d**) small black spots on leaves (diameter: <2 mm); (**e**) gray-brown necrotic lesions on leaves (diameter: 5–15 mm); (**f**) brown banded spots on leaves (diameter: 5–25 mm).

### 3.2. Isolation, Collection and Identification of Walnut Anthracnose

After isolation and purification of walnut anthracnose samples collected in the field, a total of 377 strains were obtained. Combined with the colony characteristics and amplified ITS sequences, 61 representative strains isolated from different districts and walnut parts were selected to construct phylogenetic tree. They fell into three categories, 59 of the isolates were from the *Gloeosporioides* species complex, one isolate (XGZ3011) was from the *Magnum* species complex and one isolate (XGZ3021) was from the *Orchidearum* species complex (Figure S1).

### 3.3. Phylogenetic Analysis

Based on the above results, the ACT, CAL, CHS-1, GAPDH, ITS and TUB2 genes of each isolate were amplified and uploaded to NCBI (Table S4). A Bayesian phylogenetic tree was constructed for the above 59 *Gloeosporioides* species complex isolates and 30 model strains using the six amplified genes, which were concatenated in the order of ACT-CAL-CHS1-GAPDH-ITS-TUB2 (2851 bases). The 59 isolates were classified into four different species: Twelve strains belonged to *C. aenigma*, nine to *C. fructicola*, 19 to *C. gloeosporioides* and 19 to *C. siamense* (Figure 3).

Using the same approach as above, another Bayesian phylogenetic tree was constructed for the isolated *C. magnum* and *C. orchidearum* species complex, along with 30 model strains from NCBI (Figure 4). The same six genes were concatenated in the order of ACT-CAL-CHS-1-GAPDH-ITS-TUB2, resulting in a sequence of approximately 1875 bases. The novel strain XGZ3011 isolated in this study belonged to the *C. magnum* species complex and was classified as a strain of *C. liaoningense* (Figure 4). At the same time, novel strain XGZ3021 belonged to the *C. orchidearum* species complex and was classified as a strain of *C. sojae*.

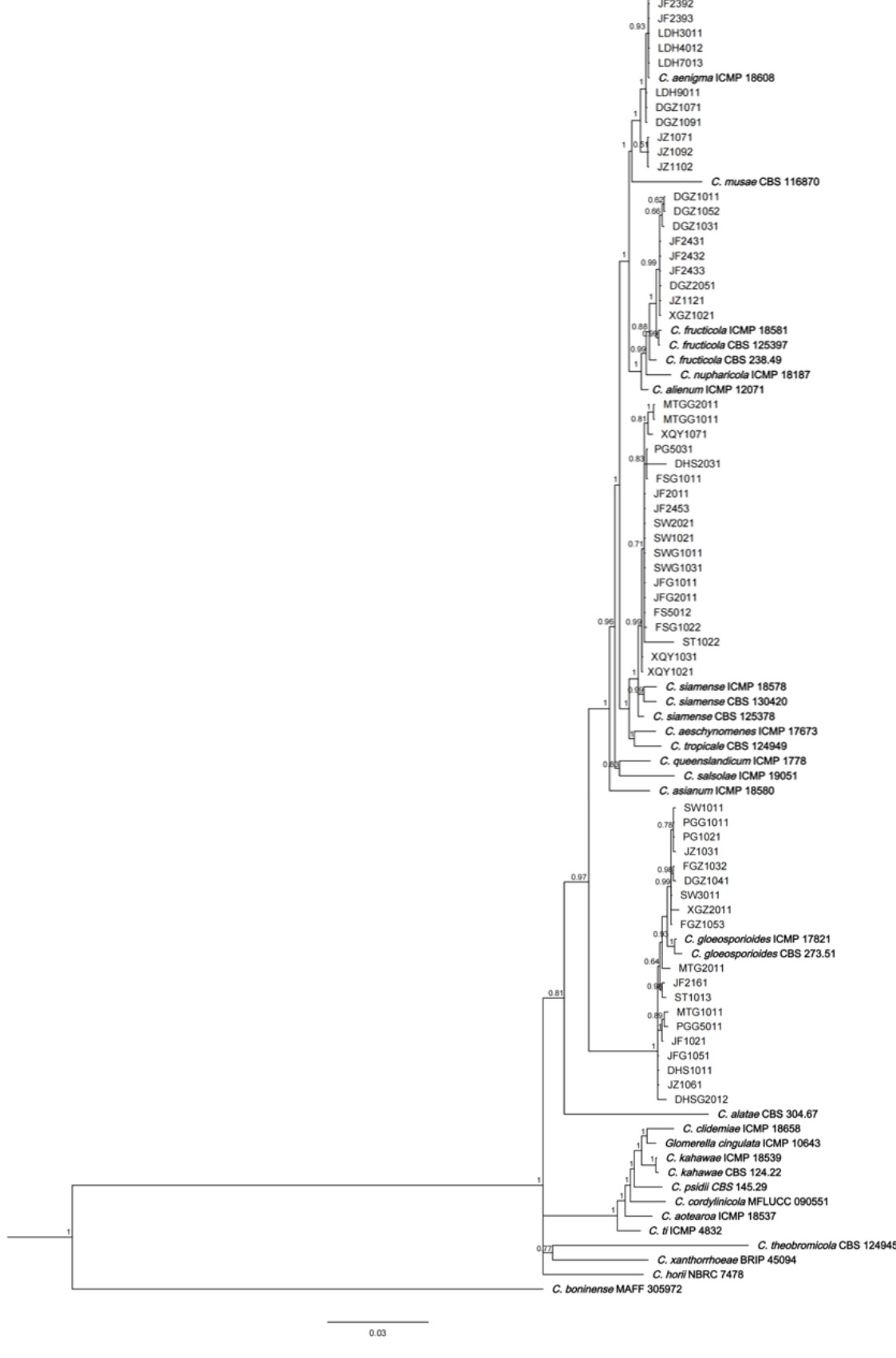

**Figure 3.** Phylogenetic tree based on the ACT-CAL-CHS-1-GAPDH-ITS-TUB2 sequences of 59 isolates and 30 model strains of the *C. gloeosporioides* species complex. Model strains are shown in bold. The scale bar shows a change of 0.03 for each site.

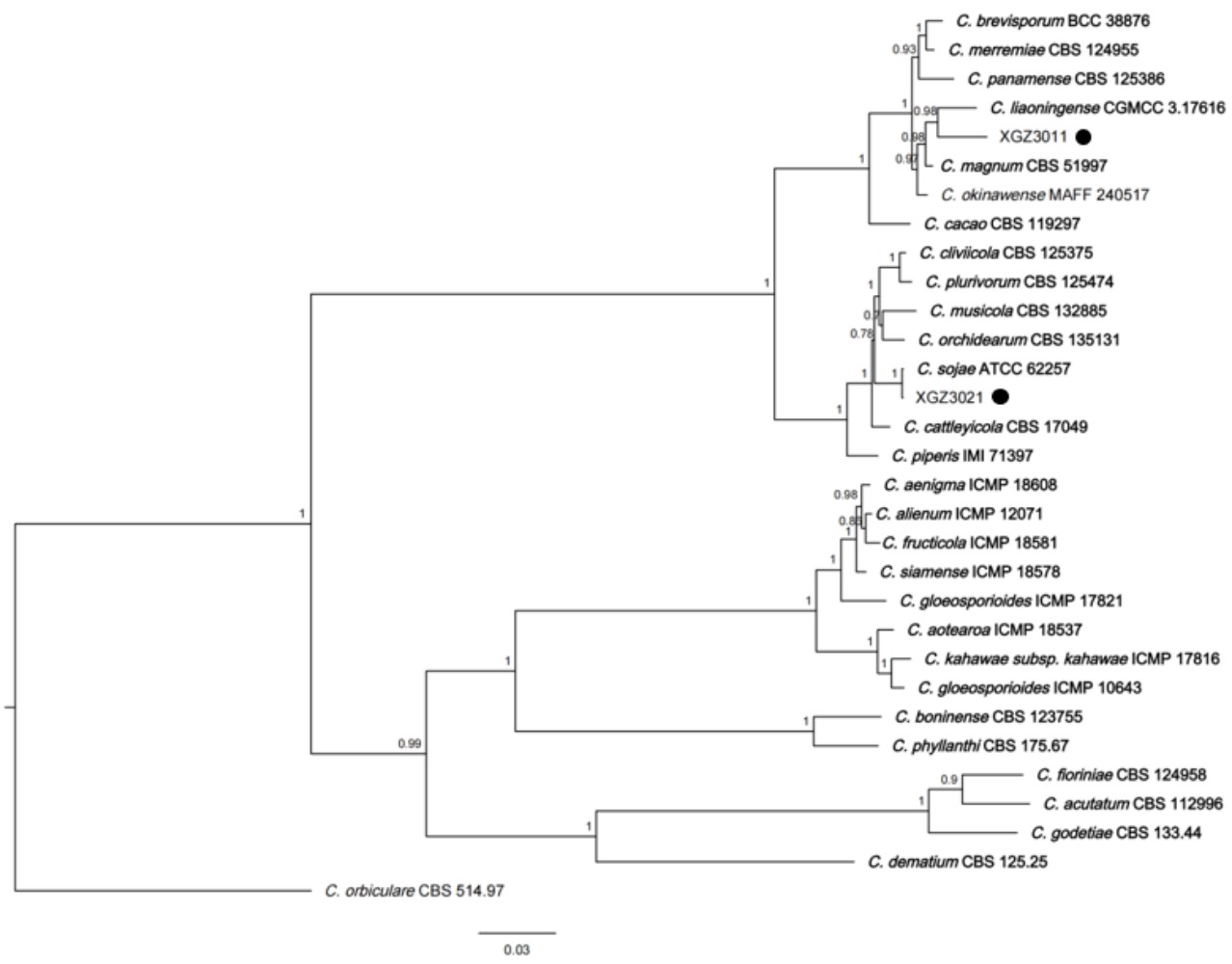

**Figure 4.** Phylogenetic tree based on the ACT-CAL-CHS-1-GAPDH-ITS-TUB2 sequences from 31 *Colletotrichum* species in this study. Model strains are shown in bold. Black circles indicate novel strains isolated in this study. The scale bar shows a change of 0.03 for each site.

### 3.4. Morphological Characteristics

Six species from the anthracnose isolates were purified and cultured on PDA plates for observation. Four of them, *C. aenigma*, *C. fructicola*, *C. gloeosporioides* and *C. siamense*, formed similar mycelia. Those colonies were 72–85 mm in diameter and flat with round edges after 5 days. Aerial mycelia were abundant, cottony, pale or grey-white in colour. In contrast, *C. fructicola* and *C. siamense* were white-green with scattering concentric whorls, *C. aenigma* formed a ring of black concentric circles and *C. gloeosporioides* grew in irregular dark concentric whorls. Conidial stacks were sticky and pink to orange, and conidia were long ovoids, transparent and smooth, with an average size of 15.39–19.74 µm × 5.10–5.97 µm. As for *C. liaoningense* and *C. sojae*, colonies were 70 mm and 80 mm in diameter, respectively, and flat with intact edges after 5 days. There were relatively few aerial mycelia, but they were dense and white. In reverse, *C. liaoningense* were entirely dark green to black with scattered concentric circles and white edges. *C. sojae* were light pink to orange, with concentric circles that were darker in the periphery. Conidial stacks were scarce, rod-shaped and transparent, with an average size of 19.43–21.49 µm × 4.73–5.21 µm and 18.51–25.79 µm × 4.95–5.93 µm (Figure 5).

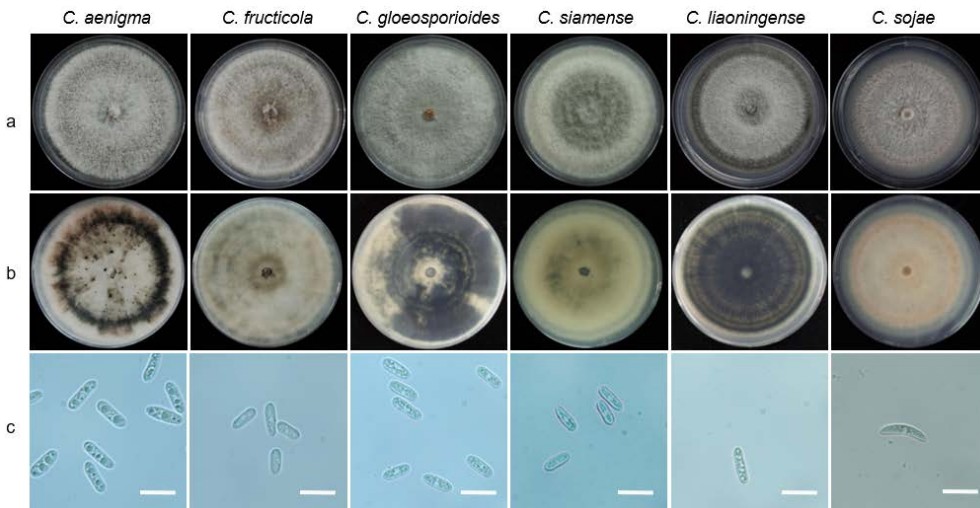

**Figure 5.** Morphological characteristics of *Colletotrichum* isolates from walnut anthracnose cultured on PDA for 5 days. (**a**) and (**b**), upper and reverse view of colonies. (**c**), conidia. Scale bar = 20 μm.

### 3.5. Pathogen Diversity Analysis

In this study, 377 walnut anthracnose strains, including six species of *Colletotrichum* were isolated from seven districts of Beijing (Table 3). Among them, *C. gloeosporioides* was the dominant species with an isolation frequency of 43.24% and 163 strains distributed in seven districts. *C. siamense* was the second most common with an isolation frequency of 41.38% and 156 strains scattered in five districts. *C. liaoningense* and *C. sojae* had the lowest isolation frequency of 0.8% each. Up to 138 strains were collected in Haidian, making it the most diverse district, whereas the Changping district had the fewest strains and was the least diverse. Eighty-one strains, including *C. liaoningense* and *C. sojae*, were isolated from the Pinggu district (Table 3). In terms of plant tissues, 82 strains consisting of two species, *C. gloeosporioides* and *C. siamense*, were isolated from walnut fruits, whereas the other 295 strains, consisting of six species, were isolated from walnut leaves.

**Table 3.** Regional diversity of *Colletotrichum* spp. of walnut anthracnose in Beijing.

| District | Orchard | Longitude and Latitude | C. a | C. f | C. g | C. si | C. l | C. so | Total |
|---|---|---|---|---|---|---|---|---|---|
| Changping | Shatuo village | N: 40°21′43.7″ E: 116°42′75.3″ | | | 3 | 3 | | | 6 |
| Fangshan | Shawo village | N: 39°76′37.0″ E: 116°04′54.7″ | | | 24 | 33 | | | 57 |
| | Fangshan | N: 39°81′95.2″ E: 116°04′09.0″ | | | | 10 | | | 10 |
| Haidian | Jiufeng mountain | N: 40°05′71.3″ E: 116°12′69.8″ | 3 | 3 | 20 | 68 | | | 94 |
| | Xueqinglu road | N: 40°01′61.7″ E: 116°35′95.5″ | | | | 38 | | | 38 |
| Huairou | Liuduhe village | N: 40°39′08.0″ E: 116°54′34.3″ | 11 | | 3 | | | | 14 |
| | Denggezhuang village | N: 40°43′47.6″ E: 116°70′89.8″ | 7 | 12 | 3 | | | | 22 |
| Mentougou | Mentougou | N: 39°96′01.4″ E: 115°70′75.8″ | | | 9 | 2 | | | 11 |
| Miyun | Fengezhuang village | N: 40°45′30.9″ E: 116°79′20.4″ | | | 15 | | | | 15 |
| | Jizhuang village | N: 40°39′68.7″ E: 116°82′64.0″ | 12 | 3 | 14 | | | | 29 |
| Pinggu | Pinggu | N: 40°19′58.7″ E: 117°12′29.1″ | | | 65 | 1 | | | 66 |
| | Dahuashan town | N: 40°27′91.0″ E: 117°08′65.9″ | | | 4 | 1 | | | 5 |
| | Xiagezhuang town | N: 40°14′28.9″ E: 117°14′87.4″ | | 1 | 3 | | 3 | 3 | 10 |
| Total | | | 33 | 19 | 163 | 156 | 3 | 3 | 377 |

### 3.6. Effects of Temperature on Pathogen Growth

Fourteen isolates were selected as follows for this temperature experiment: *C. aenigma* (DGZ1071, LDH9011 and JF2391), *C. fructicola* (DGZ1052, JF2432 and JZ1121), *C. gloeosporioides* (FGZ1011, JF1021 and PG1021), *C. liaoningense* (XGZ3011), *C. siamense* (JF2011, SWG1011 and XQY1021) and *C. sojae* (XGZ3021). The optimum growth temperature was 28 °C for all six species (Figure 6). At this temperature, *C. aenigma* and *C. gloeosporioides* had the most vigorous growth, with average colony growth rates of 15.59 and 15.22 mm/d, respectively. These were followed by *C. fructicola*, *C. sojae* and *C. siamense*, which had average growth rates of 14.86, 14.75 and 13.58 mm/d. *C. liaoningense* had the slowest growth rate, 12.8 mm/d.

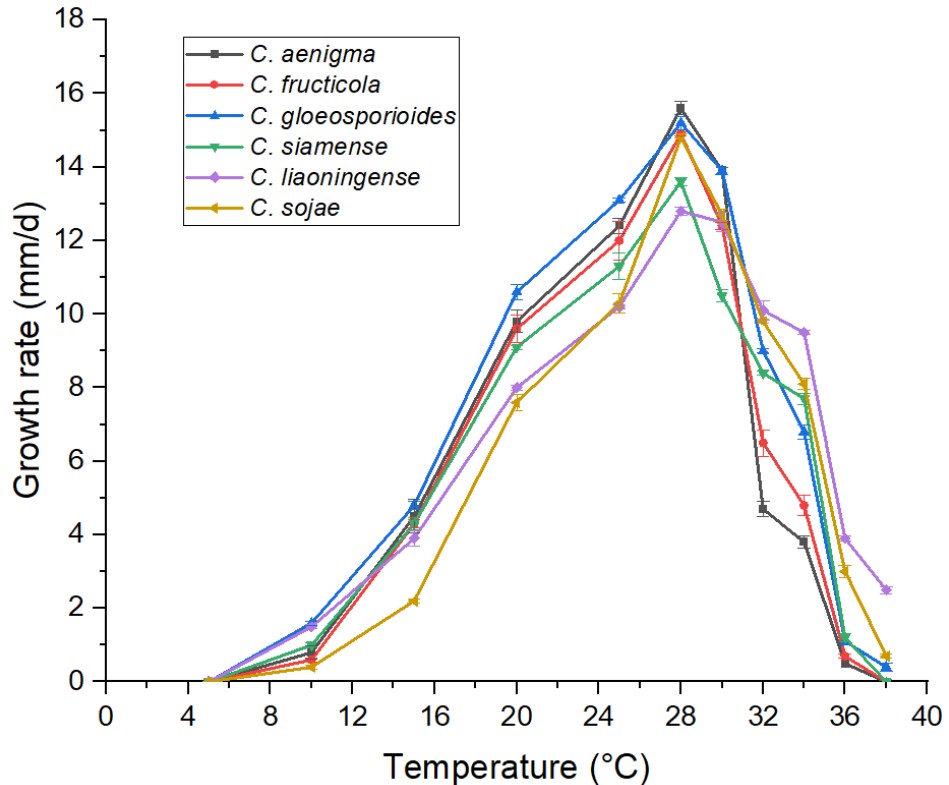

**Figure 6.** Colony growth rate of walnut anthracnose under different temperatures for four days.

Different strains could stand various temperatures. Results showed *C. gloeosporioides*, *C. liaoningense* and *C. sojae* could grow at up to 38 °C. At 38 °C, *C. liaoningense* showed the strongest temperature resistance with an average growth rate of 2.5 mm/d, followed by *C. sojae* and *C. gloeosporioides* at 0.7 and 0.43 mm/d. *C. aenigma*, *C. fructicola* and *C. siamense* showed poorer resistance to high temperature; mycelial growth was not observed at 36 °C. In addition, the low temperature resistance trial showed no mycelial growth at 5 °C for all six species.

### 3.7. Pathogenicity Test

Walnut leaves were inoculated with 14 isolates from *Gloeosporioides*, *Magnum* and *Orchidearum* species. Infection occurred at the inoculation sites on detached leaves, and symptoms resembled the symptoms of anthracnose disease in the field (Figure 7).

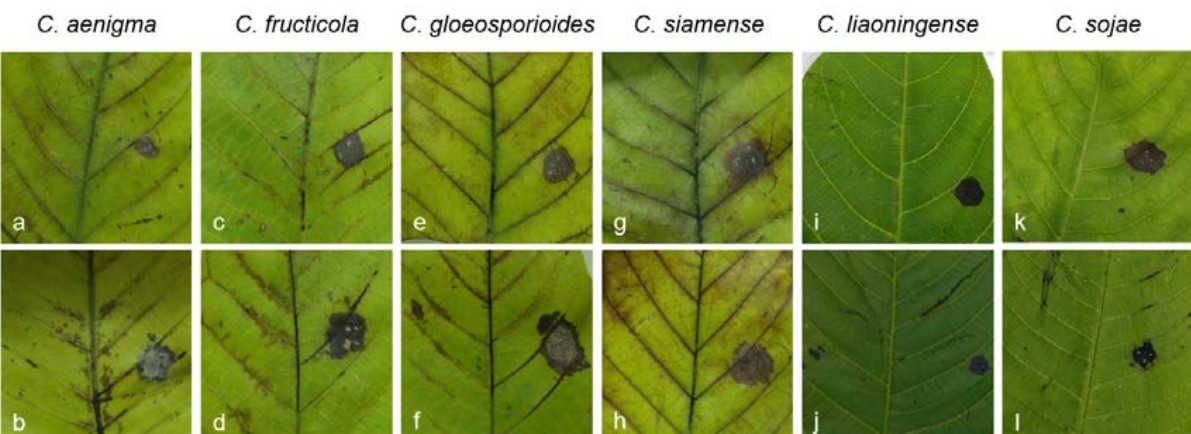

**Figure 7.** Pathogenicity of *Colletotrichum* species induced by inoculation of spore suspensions under wounded conditions. (**a**) DGZ1071. (**b**) JF2391. (**c**) JZ1121. (**d**) XGZ1021. (**e**) JF1021. (**f**) PG1021. (**g**) JF2011. (**h**) XQY1021. (**i**) and (**j**) XGZ3011. (**k**) and (**l**) XGZ3021.

All stab wound inoculations from all isolates caused anthracnose symptoms on walnut leaves after 4 days, whereas the leaves mock-inoculated in the control group developed no symptoms. The typical symptoms appeared as approximate circular or irregular spots around the water-soaked inoculation sites, and they changed from brown to black. Once black, the lesions expanded and turned into necrotic areas. The fungi were reisolated and Koch's principles were completed. Results indicated that all 14 selected isolates were walnut anthracnose pathogens. There were significant virulence differences ($p < 0.01$) among 14 isolates. *C. gloeosporioides* strains JF1021, PG1021 and JZ1031 produced lesions with the greatest spread and highest virulence, with average lesion sizes of 12.33 mm ± 0.29 mm. *C. fructicola* (XGZ1021, JF1121, DGZ1052) and *C. siamense* (JF2011, XQY1021, SWG1011) were the second with average growth diameters of 10.42 mm ± 1.88 mm and 10.17 mm ± 1.23 mm, respectively. Furthermore, *C. liaoningense* (XGZ3011) and *C. sojae* (XGZ3021) showed weak infectivity, with average lesion diameters of 7.83 mm ± 1.42 mm and 6.83 mm ± 2.01 mm. *C. aenigma* strains (DGZ1071, JF2391, LDH9011) had the smallest average lesion size of 5.50 mm ± 0.87 mm (Table 4). Taken together, all strains showed different virulence patterns between species.

**Table 4.** Pathogenicity of different strains inoculated on walnut leaves after 4 days.

| Species | | Spot Diameter (mm) [x] |
|---|---|---|
| Gloeosporioides | *C. aenigma* | 5.50 ± 0.87 [a] |
| | *C. fructicola* | 10.42 ± 1.88 [bc] |
| | *C. gloeosporioides* | 12.33 ± 0.29 [c] |
| | *C. siamense* | 10.17 ± 1.23 [bc] |
| Magnum | *C. liaoningense* | 7.83 ± 1.42 [ab] |
| Orchidearum | *C. sojae* | 6.83 ± 2.01 [a] |

[x] Spot diameter is reported as the mean ± SD. Values followed by the same letter are not significantly different according to a one-way ANOVA test with LSD ($p = 0.05$).

### 3.8. In Vitro Sensitivity of Isolates to Seven Fungicides

Results showed that the inhibitory effects of the seven studied fungicides on mycelial growth of each pathogen was significantly different (Table 5 and Table S5). Overall, SYP-14288 and prochloraz had the strongest inhibitory effects with average $EC_{50}$ values from 0.01–0.05 µg/mL, followed by fluazinam, difenoconazole and epoxiconazole, which had average $EC_{50}$ values from 0.05–0.36 µg/mL. The last two fungicides, tebuconazole and mefentrifluconazole, showed poor inhibitory effects, with average $EC_{50}$ values from 0.20–1.5 µg/mL.

**Table 5.** Sensitivity of *Colletotrichum* species to seven fungicides.

| Fungicides | EC$_{50}$ (µg/mL) [x] | | | | | |
|---|---|---|---|---|---|---|
| | *C. aenigma* | *C. fructicola* | *C. gloeosporioides* | *C. siamense* | *C. liaoningense* | *C. sojae* |
| Difenoconazole | 0.1850 ± 0.0775 [A] | 0.0945 ± 0.0251 [AB] | 0.1350 ± 0.0520 [A] | 0.1030 ± 0.0314 [AB] | 0.1308 ± 0.0309 [AB] | 0.2333 ± 0.0268 [B] |
| Fluazinam | 0.0676 ± 0.0198 [A] | 0.0535 ± 0.0135 [A] | 0.0590 ± 0.0117 [A] | 0.0548 ± 0.0181 [AB] | 0.1082 ± 0.0276 [AB] | 0.2570 ± 0.0452 [B] |
| Epoxiconazole | 0.3442 ± 0.1029 [A] | 0.2914 ± 0.0234 [B] | 0.3554 ± 0.1488 [B] | 0.1590 ± 0.0365 [B] | 0.2288 ± 0.0523 [AB] | 0.6782 ± 0.2494 [C] |
| Mefentriflucona-zole | 1.0523 ± 0.5442 [B] | 0.9210 ± 0.3815 [C] | 1.1524 ± 0.3415 [D] | 0.8433 ± 0.3192 [C] | 0.5701 ± 0.1733 [B] | 1.4666 ± 0.4278 [D] |
| Prochloraz | 0.0269 ± 0.0071 [A] | 0.0198 ± 0.0042 [A] | 0.0210 ± 0.0057 [A] | 0.0179 ± 0.0042 [A] | 0.0154 ± 0.0056 [A] | 0.0514 ± 0.0214 [A] |
| SYP-14288 | 0.0107 ± 0.0010 [A] | 0.0096 ± 0.0026 [A] | 0.0175 ± 0.0083 [A] | 0.0187 ± 0.0051 [A] | 0.0227 ± 0.0095 [A] | 0.0414 ± 0.0132 [A] |
| Tebuconazole | 0.8308 ± 0.5358 [B] | 0.2135 ± 0.1288 [AB] | 0.6698 ± 0.3985 [C] | 0.7244 ± 0.1216 [C] | 0.2516 ± 0.0148 [AB] | 0.6900 ± 0.3582 [C] |

[x] Mean ± SD (standard deviation); values followed by the same letters are not significantly different based on a one-way ANOVA with LSD test ($p$ = 0.05).

## 4. Discussion

At present, fungicide treatment is the most effective way to prevent and control anthracnose, but over reliance on fungicides has caused fungicide resistance that has led to breakthrough infections and severe loss. Meanwhile, none of the fungicides have been registered on the China Pesticide Information Network (CPIN) (http://www.chinapesticide.org.cn/, accessed on 8 May 2020) to control walnut anthracnose. Therefore, it is crucial to select effective fungicides for walnut anthracnose. Fungicide efficacy was studied in other regions of China. In one study, Liu et al. showed that the alternate use of prochloraz, tebuconazole, triadimefon, mancozeb and iprodione during different periods of anthracnose development was beneficial for controlling the disease on walnuts [45]. Studies from Wang et al. and Fu et al. showed that prochloraz effectively inhibited the growth of walnut anthracnose [46,47]. Through sensitivity tests of eight fungicides, Zhang et al. found that tebuconazole had the most obvious inhibitory effect on walnut anthracnose [48]. In addition, Wang et al. suggested that 25% pyraclostrobin SC had the best efficacy for controlling walnut anthracnose [49]. Meng et al. showed that out of eight fungicides, prochloraz and fludioxonil were the most potent against nine anthracnose pathogens [50]. However, these tests were not conducted for walnut anthracnose incidences in the districts of Beijing.

Fungicide sensitivity results indicated the sensitivities of each species causing walnut anthracnose to fungicides showed partial differences. Hence, based on the average EC$_{50}$ values of each species combined with dominant species from different regions, we presented a series of strategies for the targeted use of fungicides. Undoubtedly, prochloraz and SYP-14288 had the most obvious inhibitory effect on all six species, and could be used as the primary fungicide to control walnut anthracnose. However, long-term use of the most effective fungicides may lead to the development of fungicide resistance. Therefore, the selection of alternative fungicides is still critical.

In recent years, walnut cultivation area has become extensive, covering several regions in Beijing. However, walnut anthracnose occurs to varying degrees in different regions because of location, soil and cultivation conditions. Therefore, it is crucial to investigate walnut anthracnose in each cultivation regions to clarify the distribution of the pathogen strains. Our regional diversity results indicated that only *C. gloeosporioides* and *C. siamense* were isolated from Changping, Fangshan and Mentougou districts. The dominant species was *C. siamense* in Haidian district, whereas *C. aenigma* was predominant in Huairou and Miyun districts. *C. gloeosporioides* was the dominant species in the Pinggu district, and *C. liaoningense* and *C. sojae* were also obtained from the Pinggu district. Fluazinam was the best inhibitor of *C. aenigma*, *C. gloeosporioides* and *C. siamense*, and thus could be used as an alternative fungicide in Changping, Fangshan, Miyun and Mentougou districts. Difenoconazole was most effective against *C. fructicola* and *C. siamense*, and thus could be used in Haidian and Huairou districts. Epoxiconazole and tebuconazole showed inhibition to *C. liaoningense* and *C. sojae*, and could be used as an alternative in the Pinggu district. However, these fungicides need to be tested in field studies to verify whether these fungicides are effective at controlling walnut anthracnose.

Fungicide resistance is mainly caused by natural genetic variations that reduce sensitivity to fungicide. Even if these variations occur at low frequency, prolonged use of the same fungicide easily leads to the development of drug resistance in pathogens, causing a decline in the effectiveness of prevention. Currently, anthracnose is classified by the Fungicide Resistance Action Committee (FRAC) (https://www.frac.info/, accessed on 26 April 2020) as having a moderate risk of resistance. It has been reported that grape ripe rot has developed decreased susceptibility to carbendazim and thiophanate-methyl [51,52]. *C. gloeosporioides* isolated from mango, avocado, apple, banana, etc., also appeared to develop strains resistant to these fungicides [53,54]. *C. graminicola* and *C. gloeosporioides* have developed resistance to QoI fungicides, which are classified as having a high resistance risk [55,56]. DMI fungicides are classified as having a moderate risk for development of fungal resistance by FRAC, but DMI-resistant strains have been found on many crops. Strains that cause brown spot disease that are highly resistant to DMI have been isolated from sugar beet, and highly resistant strains of powdery mildew have been isolated from wheat [57]. Anthracnose strains isolated from grapes in the Zhejiang and Yunnan provinces of China have shown resistance to tebuconazole, myclobutanil and diniconazole [58].

## 5. Conclusions

In this study, field investigations were conducted to identify the pathogens causing walnut anthracnose in major walnut plantations in Beijing, and two novel pathogens were found. For comprehensive prevention of walnut anthracnose, this study recommends multiple fungicides that focus on the dominant pathogens in seven regions of Beijing.

**Supplementary Materials:** The following supporting information can be downloaded at: https://www.mdpi.com/article/10.3390/agronomy13010214/s1, Figure S1: Phylogenetic tree based on ITS sequences from 74 *Colletotrichum* isolates of walnut anthracnose obtained in this study; Table S1: Grading standard for diseased degree on walnut leaves; Table S2: Grading standard for diseased degree on walnut fruits; Table S3: Model strains and sequences of *Colletotrichum* used in this study for phylogenetic analysis; Table S4: List of 61 representative isolates of different *Colletotrichum* spp. collected from walnut in Beijing; Table S5: Determination of sensitivity of walnut anthracnose to seven fungicides.

**Author Contributions:** Conceptualization, J.C. and Y.B.; methodology, Q.C.; software, H.Z.; validation, Z.L., J.S. and Y.Y.; formal analysis, J.C. and F.L.; investigation, J.C.; resources, Q.C.; data curation, H.Z.; writing—original draft preparation, F.L.; writing—review ands editing, Y.B.; visualization, Y.B.; supervision, Y.B.; project administration, Q.C.; funding acquisition, Y.B. All authors have read and agreed to the published version of the manuscript.

**Funding:** This research was funded by the National Key Research and Development Program of China (2017YFD0201601-6); the National Natural Science Foundation of China (31501671); Beijing University of Agriculture graduate education development program (2021YJS037).

**Data Availability Statement:** The data presented in this study are available in Figure 6, Tables 2–5, Tables S4 and S5.

**Acknowledgments:** The authors thank the Shenyang Research Institute of Chemical Industry, China, and China Agricultural University for providing the fungicides used in this study.

**Conflicts of Interest:** The authors declare no conflict of interest.

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
