# Peer review of "Identification, Pathogenicity, and Sensitivity to Fungicide of Colletotrichum Species That Causes Walnut Anthracnose in Beijing"

_agronomy, doi:10.3390/agronomy13010214_

Round 1

Reviewer 1 Report

The research (Identification, pathogenicity, and sensitivity to fungicide of Colletotrichum species that cause walnut anthracnose in Beijing) investigates walnut anthracnose incidences in seven districts of Beijing. They identified the dominant walnut anthracnose Colletotrichum species, their pathogenicity as well as their sensitivity to different fungicides. Although the manuscript idea is good, there are major modifications needed throughout the manuscript:

 1.      In the method part:

- The site locations of the collected sample areas and the field investigations area need to include using GPS coordinates.

- The environmental conditions of the collected sample areas and the field investigations area must include.

- Authors need to include information about the collected sample (30 walnut trees) like the age of the tree, length, size,…. etc.

- In the morphological observation why did the authors take the spore suspension only for the examination, however, they could scratch the whole fungus from the PDA plates and examine it???

- The authors write in the temperature effect (According to the results of phylogenetic analysis, isolates from different regions and species were selected to determine the effect of temperature on colony growth) why did they connect this effect with the phylogenetic analysis??. They even not include the temperatures of the collected sample areas.

- In the pathogenicity and virulence on leaves; authors need to add the technique reference.  

- It would prefer to add a scanning electron microscope of the anthracnose.

3.      In the results part;

-          In the field investigations and description of symptoms, there is no information’s about the field condition, temperature, humidity, .etc which is unacceptable.

- The title 3.2. Isolation, collection, and identification …… of what? it needs to be more clear.

- In line 237; after isolation and purification of walnut anthracnose samples collected in the field, a total of 377 strains were obtained. Combined with the colony characteristics, 61 representative strains isolated from different districts and walnut parts were selected to amplify the ITS sequence. The 61 strains were isolated from which areas and why did they choose these isolates for ITS sequence?

- Phylogenetic tree figures need to be more clear and in good resolution.

- In Figure 5; Colletotrichum needs to be in italics.

- In Temperature effect on pathogen growth, the authors take the results after 4 days or for 4 days as mentioned?  

- In fungicide results, the authors must add the results of the different concentrations (even as supplemented data) not EC50 (μg/mL) only.

4.     The discussion part is poor and needs more new informations and references.

5. The manuscript needs English language and grammar revision.

Reviewer 2 Report

Walnut anthracnose is a significant disease issue in different regions of China. The researchers observed the incidence of anthracnose in walnut trees in 7 districts of Beijing. They isolated 377 strains of Colletotrichum species and performed MLST to identify species. They then performed pathogenicity assays, as well as fungicide sensitivity assays on the various isolates. 

Overall, I think the study has scientific merit and was carried out effectively. The authors obtained a significant amount of data that will be meaningful to the scientific community. There are major changes that I would recommend prior to publication. 

Introduction: The introduction is wordy and requires technical improvements. Given the focus on how the acquired isolates fall within the different species complex, it would be informative to include more information on the classification and nomenclature of Colletotrichum species. There was some attempt made in the discussion section, but it would be better served in the introduction. Likewise, the authors have provided a lot of information on different fungicide uses for anthracnose in the introduction that should be included in the discussion section. Some information on the economic impact of anthracnose on walnuts in China would be informative. 

Materials and Methods:

Plant collection and field investigations: This section is confusing and needs additional detail. I would suggest a more in-depth description of the rating system that was used. Was this rating previously published? The supplementary table indicates that the ratings were performed on a 1-5 scale, but the results have actual percentage values. It is important to note that categorical scales (1-5) cannot be averaged. Is that what was done for the disease index? Were 30 trees from each district observed or 30 trees across all 7 districts?

DNA extraction, amplification, and sequencing: Was the DNA quantified before PCR amplification? If so, how much DNA was used for each reaction? Were single amplicon bands obtained?

Phylogenetic analysis: How was the sequence quality? Were sequences trimmed prior to collation?

Pathogenicity and virulence on leaves: Were compound leaves or single leaflets excised from the tree? The inoculations "in vitro" is confusing since the pathogen is being inoculated onto live tissue (in vivo). This section needs to be clarified. 

In vitro sensitivity of isolates to seven fungicides: How many isolates were tested? Were several isolates of a single species tested or one representative of each species? Why was DMSO used as a control? It's not reported in the results section. I am assuming DMSO was used as the positive control; what about a negative control (no additive)?

Results:

Field investigations and description of symptoms: (Line 224-225) Is the average disease severity of all trees assessed? Figure 1 could be improved. The resolution is poor. I would suggest separate panels of disease symptoms on leaves and fruit, respectively. There is a difference between disease incidence and disease severity. Incidence would indicate, for example, the total number of leaves affected per the total number of leaves observed. Severity would indicate the percent disease coverage of diseased lesions on leaves. If the authors are reporting the incidence, then that needs to be described in the methods section. 

Phylogenetic analysis: This section is confusing. It would be easier to understand with a better background on the species complexes.

Pathogen diversity analysis: In the earlier section, it says that 61 representative isolates were selected for MLST. These results describe the species' incidence of all 377 isolates. How were the remaining isolates identified? Morphological characteristics?

Figure 4: Only shows 2 sequences from isolates in the study, not 27. 

Figure 7: Where are the negative controls?

Lines 324-337: Lesion measurements need to be described in materials and methods.

In vitro sensitivity of isolates to seven fungicides: Where are the results of the controls?

Discussion: Lines 381-333: mention Alternaria disease that was not previously discussed. This section feels like a repetition of the results. There is some meaningful dialog on fungicide usage. Specific fungicides pose a greater risk for the development of fungicide resistance than others. Fungicide resistance is not uniform. That may be a good discussion point. 

Round 2

Reviewer 1 Report

The authors did a good job of enhancing the manuscript and I recommended accepting it after English editing.

Reviewer 2 Report

The authors have made significant improvements to the manuscript, specifically the introductory and discussion sections. Some minor grammatical corrections could be made to the English writing. I have two remaining comments that I believe should be addressed before publication approval.

1. Plant collection and field investigations: How many total fruit or leaves were observed on each tree? Be explicit. 

2. Figure 7: I understand the desire to preserve the resolution of each image but having a panel showing the negative control is absolutely critical. The detached leaflets were incubated for 5-days, at which point natural senescence and degradation of tissue will be observed. From the images provided, we see darkened mid-ribs and leaf veins. Is that from the inoculation? I would not assume so, but a negative control can help the reader to clearly differentiate between what is normal, "healthy" tissue vs. diseased tissue. 
